# A multicenter, randomized, open-label, controlled trial to evaluate the efficacy and tolerability of hydroxychloroquine and a retrospective study in adult patients with mild to moderate coronavirus disease 2019 (COVID-19)

Cheng-Pin Chen[1,2]☯, Yi-Chun Lin[1,3]☯, Tsung-Chia Chen[4], Ting-Yu Tseng[4], Hon-Lai Wong[5], Cheng-Yu Kuo[6], Wu-Pu Lin[7], Sz-Rung Huang[8], Wei-Yao Wang[9], Jia-Hung Liao[10], Chung-Shin Liao[11], Yuan-Pin Hung[12], Tse-Hung Lin[13], Tz-Yan Chang[13], Chin-Fu Hsiao[14], Yi-Wen Huang[13,15], Wei-Sheng Chung[4,16,17], Chien-Yu Cheng[1,18], Shu-Hsing Cheng[1,19]*, on behalf of the Taiwan HCQ Study Group¶

1 Department of Infectious Diseases, Taoyuan General Hospital, Ministry of Health and Welfare, Taoyuan, Taiwan, 2 Institute of Clinical Medicine, National Yang-Ming University, Taipei, Taiwan, 3 Graduate Institute of Clinical Medicine, Taipei Medical University, Taipei, Taiwan, 4 Department of Internal Medicine, Taichung Hospital, Ministry of Health and Welfare, Taichung, Taiwan, 5 Department of Internal Medicine, Keelung Hospital, Ministry of Health and Welfare, Keelung City, Taiwan, 6 Department of Internal Medicine, Pingtung Hospital, Ministry of Health and Welfare, Pingtung, Taiwan, 7 Department of Internal Medicine, Taipei Hospital, Ministry of Health and Welfare, New Taipei City, Taiwan, 8 Department of Internal Medicine, Miaoli General Hospital, Ministry of Health and Welfare, Miaoli, Taiwan, 9 Department of Internal Medicine, Feng Yuan Hospital, Ministry of Health and Welfare, Taichung, Taiwan, 10 Department of Internal Medicine, Nantou Hospital, Ministry of Health and Welfare, Nantou, Taiwan, 11 Department of Internal Medicine, Chia Yi Hospital, Ministry of Health and Welfare, Chiayi, Taiwan, 12 Department of Internal Medicine, Tainan Hospital, Ministry of Health and Welfare, Tainan City, Taiwan, 13 Department of Internal Medicine, Chang Hua Hospital, Ministry of Health and Welfare, Changhua, Taiwan, 14 Institute of Population Health Sciences, National Health Research Institutes, Zhunan, Taiwan, 15 Institute of Medicine, Chung Shan Medical University, Taichung, Taiwan, 16 Department of Health Service Administration, China Medical University, Taichung, Taiwan, 17 Department of Healthcare Administration, Central Taiwan University of Science and Technology, Taichung, Taiwan, 18 School of Public Health, National Yang-Ming University, Taipei, Taiwan, 19 School of Public Health, Taipei Medical University, Taipei, Taiwan

☯ These authors contributed equally to this work.
¶ Membership of the Taiwan HCQ Study Group is listed in the Acknowledgments.
* shcheng@mail.tygh.gov.tw

## Abstract

### Objective

In this study, we evaluated the efficacy of hydroxychloroquine (HCQ) against coronavirus disease 2019 (COVID-19) via a randomized controlled trial (RCT) and a retrospective study.

### Methods

Subjects admitted to 11 designated public hospitals in Taiwan between April 1 and May 31, 2020, with COVID-19 diagnosis confirmed by pharyngeal real-time RT-PCR for SARS-CoV-2, were randomized at a 2:1 ratio and stratified by mild or moderate illness. HCQ (400 mg

**Data Availability Statement:** All data are available from the figshare database( https://doi.org/10. 6084/m9.figshare.12631403.v1).

**Funding:** SHC received research grant from the Hospital and Social Welfare Organizations Administration Commission, Ministry of Health and Welfare (https://www.hso.mohw.gov.tw/). This funding source played no role in study design or conduction, data collection, analysis or interpretation, writing of the manuscript, or decision to submit it for publication.

**Competing interests:** The authors have declared that no competing interests exist.

twice for 1 d or HCQ 200 mg twice daily for 6 days) was administered. Both the study and control group received standard of care (SOC). Pharyngeal swabs and sputum were collected every other day. The proportion and time to negative viral PCR were assessed on day 14. In the retrospective study, medical records were reviewed for patients admitted before March 31, 2020.

## Results

There were 33 and 37 cases in the RCT and retrospective study, respectively. In the RCT, the median times to negative rRT-PCR from randomization to hospital day 14 were 5 days (95% CI; 1, 9 days) and 10 days (95% CI; 2, 12 days) for the HCQ and SOC groups, respectively ($p = 0.40$). On day 14, 81.0% (17/21) and 75.0% (9/12) of the subjects in the HCQ and SOC groups, respectively, had undetected virus ($p = 0.36$). In the retrospective study, 12 (42.9%) in the HCQ group and 5 (55.6%) in the control group had negative rRT-PCR results on hospital day 14 ($p = 0.70$).

## Conclusions

Neither study demonstrated that HCQ shortened viral shedding in mild to moderate COVID-19 subjects.

## Introduction

Coronavirus disease 2019 (COVID-19) is caused by severe acute respiratory syndrome coronavirus 2 (SARS-CoV-2) and is an ongoing pandemic. The outbreak was first localized to Wuhan, Hubei Province, People's Republic of China (PRC) on December 31, 2019 [1]. On January 30, 2020, the World Health Organization (WHO) declared that the outbreak was a public health emergency of international concern and thereafter recognized it as a pandemic [2, 3]. As of June 20, 2020, more than eight million cases of COVID-19 have been reported in 187 countries and territories. More than 450,000 deaths have been associated with this infection [4]. Taiwan is a close neighbor of PRC and reported its first COVID-19 case on January 21, 2020 [5]. As of June 20, 2020, there were 446 confirmed COVID-19 cases in Taiwan. As a result of the early implementation of social distancing, hand hygiene, and face masks, Taiwan has had a low incidence of domestic COVID-19 cases [6].

There is no known effective medical treatment against COVID-19. The mechanisms of potentially efficacious antiviral agents include the inhibition of RNA-dependent RNA polymerase (remdesivir [7–9] and favipiravir [10]), protease inhibition (lopinavir/ritonavir [9, 11, 12] and ivermectin [13]), the blockade of virus-cell membrane fusion (recombinant human angiotensin-converting enzyme 2 [14] and chloroquine and hydroxychloroquine (HCQ) [8, 15], and the modulation of the human immune system (interferon [9] and interleukin-6 blockers [16, 17]).

Chloroquine phosphate is a well-known antimalarial drug that has been on the market for several decades. An *in vitro* study showed that chloroquine is effective against SARS-CoV-2 at the entry and post-entry infection stages [8]. Chloroquine may either increase endosomal pH by blocking the fusion of the virus and the host cell membrane [18] or by interfering with cell receptor glycosylation [19]. Chloroquine may also repress proinflammatory signaling and cytokine (IL-1, IL-6, and TNF) production by inhibiting lysosome activity in antigen-presenting cells [20]. Compared to chloroquine, HCQ has an additional hydroxyl group, lower toxicity, and similar antiviral efficacy.

HCQ received emergency approval by the US Food and Drug Administration (FDA) for use in the treatment of COVID-19 [21]. However, the efficacy of HCQ against SARS-CoV-2 has been highly controversial. Certain elite journals have retracted influential papers published on the efficacy of HCQ against COVID-19 [22, 23]. HCQ is widely available in Taiwan and has become a potential candidate drug therapy against COVID-19 there. Therefore, an open-label randomized controlled trial (RCT) involving multiple centers was conducted to evaluate HCQ efficacy and tolerability in adult patients with mild to moderate COVID-19. These results would be compared with the standard of care treatment (SOC) in Taiwan. In addition, a retrospective observational study was also performed.

## Methods

### Clinical trial

**Participants.** The clinical trial was conducted at 11 public hospitals in northern, central, and southern Taiwan, including Keelung Hospital, Taipei Hospital, Taoyuan General Hospital, Miaoli General Hospital, Taichung Hospital, Feng Yuan Hospital, Nantou Hospital, Chang Hua Hospital, Chia Yi Hospital, Tainan Hospital, and Pintung Hospital, affiliated with the Ministry of Health and Welfare, Taiwan, between April 1 and May 31, 2020. Enrolled patients were aged 20–79 y and confirmed positive for SARS-CoV-2 infection by real-time reverse transcription polymerase chain reaction (rRT-PCR). They provided signed informed consent to participate in this study. Upon admission, the patients were stratified into three groups: (1) mild illness without evidence of infiltration according to chest X-ray; (2) moderate illness with evidence of infiltration according to chest X-ray but neither respiratory distress nor supplemental oxygen requirement; and (3) severe illness with respiratory distress, oxygen supplementation, and evidence of infiltration according to chest X-ray. Participants in group 3, presenting with severe illness, were excluded from this study. The following patients were excluded from the trial: (a) documented history of hypersensitivity to quinine derivatives; (b) retinal disease; (c) hearing loss; (d) severe neurological or mental illness; (e) pancreatitis; (f) lung disease; (g) liver disease (alanine aminotransferase/aspartate aminotransferase > 3× the normal upper limit); (h) kidney disease (estimated glomerular filtration rate < 30 mL/min/ 1.73 $m^2$ according to MDRD or CKD-EPI); (i) hematological disease; (j) cardiac conduction abnormalities at electrocardiographic screening with long QT syndrome or QTcF interval > 450 msec for males and > 470 msec for females according to Fridericia's correction at screening; (k) known HIV infection; (l) active hepatitis B or C without concurrent treatment (positive for hepatitis B [HBsAg and HBeAg] or hepatitis C ribonucleic acid [RNA] titer > 800,000 IU/mL); (m) G6PD; (n) psychiatric disorders and alcohol/substance dependence/abuse that may jeopardize patient safety; and (o) pregnant or breast-feeding women.

**Clinical course.** COVID-19 symptoms were recorded and followed up daily. Chest X-rays, electrocardiography, and the biomarkers complete blood count, white blood cell differential count, biochemistry, prothrombin time, activated partial thromboplastin time, ferritin, highly active troponin I, C-reactive protein, and erythrocyte sedimentation rate were tested upon admission and every 4 days after enrollment.

**PCR assay.** Nasopharyngeal swab and sputum were collected every other day until patient discharge following three consecutive negative results or day 14 of the study, depending upon which criterion was met first. SARS-CoV-2 RNA was assessed by rRT-PCR using a hydrolysis probe-based system targeting genes encoding envelope (E) protein and RNA-dependent RNA polymerase (R) as previously described [24]. Negative viral RNA detection was defined as cycle thresholds (Ct) values > 38 for the E gene and negative for the R gene. The PCR assay was conducted at the National Laboratory of the Taiwan Centers for Disease Control.

**Study design.** Eligible subjects were randomly assigned by an interactive web response system in a 2:1 ratio to receive either HCQ plus standard of care (SOC) or SOC alone. They were stratified by mild or moderate illnesses within 4 days of diagnosis. The incidence of domestic cases was low and the estimated case number was 45 (30:15). The HCQ administration plan was 400 mg b.i.d. on day 1 and 200 mg b.i.d. for 6 days on days 2–7. Both study group and comparison group received standard of care comprising supportive treatment without antibiotics for subjects with mild clinical COVID-19 symptoms and with antimicrobial therapy for subjects presenting with moderate clinical COVID-19 symptoms. The treatment consisted of: (1) ceftriaxone 2 g daily for 7 days ± azithromycin 500 mg on day 1 and 250 mg on days 2–5; or (2) levofloxacin 750 mg daily for 5 d; or (3) levofloxacin 500 mg daily; or (4) moxifloxacin 400 mg daily for 7–14 days for subjects allergic to ceftriaxone or azithromycin or according to physician discretion. Oseltamivir 75 mg b.i.d. was administered for 5 days to subjects presenting with concomitant influenza A or B infection. No HCQ dose reduction, modification, or change in administration frequency was recommended during the study period. This prospective randomized controlled clinical trial was registered at https://clinicaltrials.gov/ct2/show/NCT04384380?term=Hydroxychloroquine&cond=COVID-19&cntry=TW&draw=2&rank=1(NCT04384380) (Because the appendix of the protocol was in Mandarin, the text had to be translated to English to meet QC criteria. The registration of the clinical trial was accepted on May/12/2020, after the first enrollment on Apr/1/2020. However, the authors confirm that all ongoing and related trials for this drug/intervention were registered).

**Outcome measurement.** The primary endpoint was to evaluate the time to negative rRT-PCR assessments from randomization, up to 14 days, by arm. The secondary endpoints were to evaluate the proportion of negative viral rRT-PCR on hospital day 14, the resolution of clinical symptoms (time to clinical recovery), the proportion of discharges by day 14, and the mortality rate. HCQ safety and tolerability were also evaluated.

**Statistical analysis.** Data were entered into an electronic clinical trial information management system (CTIMeS; National Health Research Institutes, Taiwan) by study coordinators and summarized with SAS® v. 9.2 (SAS Institute Inc., Cary, NC, USA). All treatment data are summarized using descriptive statistics including continuous variables (number of non-missing observations, means, standard deviations (SD), medians, minima, and maxima), categorical variables (frequencies and percentages), and time to event variables (number of non-missing observations (N), medians, minima, and maxima). The negative rRT-PCR rates between treatment and control arms were compared using Fisher's exact test and Cochran-Mantel-Haenszel (CMH) test, stratified by mild or moderate illness. The Kaplan-Meier method was used to estimate the distribution of time to a negative rRT-PCR test. The median time to a negative rRT-PCR and its 95% CI were calculated. The log-rank test with/without adjustment by disease severity was used to compare the distribution of time to negative rRT-PCR between arms. An analysis of covariance (ANCOVA), adjusted for mild/moderate diseases, and repeated measurements of analysis of variance (ANOVA) were performed to compare the area under the curve and Ct value across different time between arms. All tests were two-tailed. A $p$-value $<0.05$ was considered significant.

**Ethical statement.** The study protocol was approved by the Institutional Review Board of Taoyuan General Hospital (IRB No. TYGH109014). The study was performed according to Good Clinical Practices recommended by the Declaration of Helsinki and its amendments.

## Retrospective observational study

The study was conducted at the same 11 hospitals. Cases were aged 20–79 y and confirmed positive for SARS-CoV-2 infection by rRT-PCR between January 25 and March 31, 2020.

Medical registers were reviewed and clinical symptoms, laboratory data, and medications were recorded. Patients who had undetected virus within 2 days of hospitalization were excluded. The study protocol was approved by the Institutional Review Board of Taoyuan General Hospital (IRB No. TYGH109024).

## Results

### Clinical trial

Thirty-three cases were enrolled in the RCT (Fig 1). The mean age (SD) of the subjects was 32.9 (10.7) y. Males comprised 57.6% of all subjects. A few individuals presented with underlying chronic illnesses. The initial presentation included anosmia (51.5%), cough (48.5%; 94% of these had mild cough and needed no symptomatic treatment), ageusia (30.0%), nasal obstruction (24.2%), and sore throat (21.2%). Of these, 12.1% of the cases had pneumonia according to the X-ray images (Table 1).

Twenty-one cases were randomized to the HCQ group and 12 cases were randomized to the SOC group. However, two in the HCQ group and one in the SOC group had withdrawn consents before the first dose was administered. One (4.8%) in the HCQ group and two (16.7%) in the SOC group were concomitantly administered azithromycin. The median times to negative rRT-PCR assessment from randomization to hospital day 14 were 5 days (95% CI; 1, 9 days) for the HCQ group and 10 days (95% CI; 2, 12 days) for the SOC group (*p* = 0.40) (Fig 2; Table 2). By day 14, 81.0% (17/21) and 75.0% (9/12) of the subjects in the HCQ and

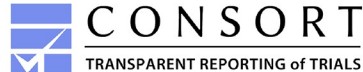
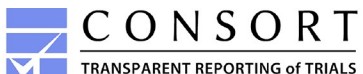
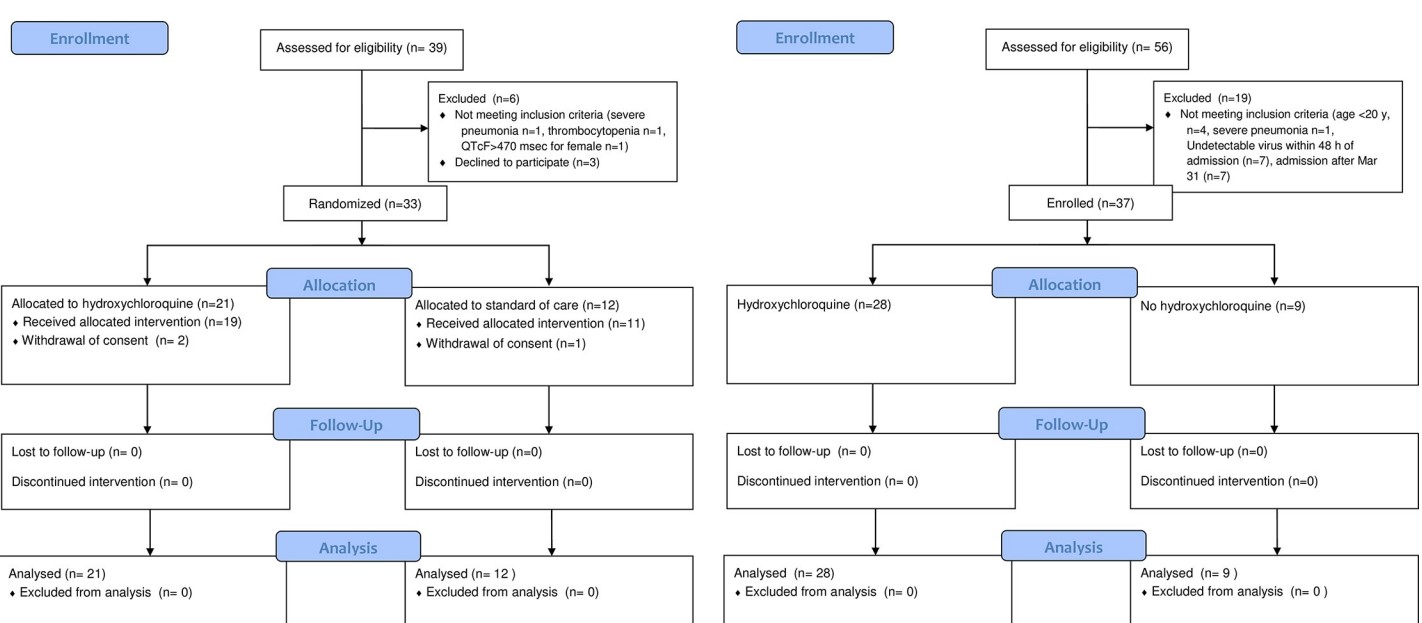

**Fig 1.** Patient disposition in the multicenter, open-label, randomized controlled trial (a) and the retrospective study (b) of hydroxychloroquine. Abbreviations: HCQ: hydroxychloroquine; SOC: standard of care.

**Table 1. Baseline demographic and clinical characteristics of participants in the multi-center, open-label, randomized clinical trial and the retrospective observational study.**

| | Randomized controlled trial | | | | | |
|---|---|---|---|---|---|---|
| | HCQ[a] | | SOC[b] | | Overall | |
| No. randomized patients | 21 | | 12 | | 33 | |
| Mean age in years (sd) | 33.0 | (12.0) | 32.8 | (8.3) | 32.9 | (10.7) |
| Median (range) | 30.0 | (22–68) | 33.5 | (22–44) | 31.0 | (22–68) |
| Male (%) | 11 | (52.4%) | 8 | (66.7%) | 19 | (57.6%) |
| Stratification | | | | | | |
| Mild (%) | 19 | (90.5%) | 10 | (83.3%) | 29 | (87.9%) |
| Moderate (%) | 2 | (9.5%) | 2 | (16.7%) | 4 | (12.1%) |
| Symptoms | | | | | | |
| Median of QTc msec (range) | 424 | (356–453) | 427.5 | (392–458) | 424 | (356–458) |
| Anosmia (%) | 11 | (52.4%) | 6 | (50%) | 17 | (51.5%) |
| Cough (%) | 9 | (42.9%) | 7 | (63.6%) | 16 | (48.5%) |
| Ageusia (%) | 4 | (19.0%) | 6 | (50%) | 10 | (30.3%) |
| Nasal obstruction (%) | 4 | (19.0%) | 4 | (33.3%) | 8 | (24.2%) |
| Sore throat (%) | 3 | (14.3%) | 4 | (33.3%) | 7 | (21.2%) |
| Shortness of breath (%) | 1 | (4.8%) | 1 | (8.3%) | 2 | (6.1%) |
| Fever (%)[c] | 1 | (4.8%) | 0 | (0%) | 1 | (3.0%) |
| | Retrospective observational study | | | | | |
| | HCQ | | Control | | Total | |
| No. patients | 28 | | 9 | | 37 | |
| Mean age (Std) | 34.3 | (14.5) | 31.3 | (18.0) | 35.8 | (14.5) |
| Median (range) | 28 | (20–66) | 44 | (21–56) | 29 | (20–66) |
| Male (%) | 14 | (50%) | 3 | (33.3%) | 17 | (45.9%) |
| Stratification | | | | | | |
| Mild (%) | 23 | (82.1%) | 6 | (66.7%) | 29 | (78.4%) |
| Moderate (%) | 5 | (17.9%) | 3 | (33.3%) | 8 | (21.6%) |
| Symptoms | | | | | | |
| Median of QTc msec (range) | NA | | NA | | NA | |
| Anosmia (%) | 8 | (28.6%) | 1 | (11.1%) | 9 | (24.3%) |
| Cough (%) | 18 | (64.3%) | 3 | (33.3%) | 21 | (56.8%) |
| Ageusia (%) | 5 | (17.9%) | 1 | (11.1%) | 6 | (16.2%) |
| Sore throat (%) | 8 | (28.6%) | 2 | (22.2%) | 10 | (27.0%) |
| Shortness of breath (%) | 0 | (0%) | 1 | (11.1%) | 1 | (2.7%) |
| Fever (%)[c] | 15 | (53.6%) | 3 | (33.3%) | 18 | (48.6%) |

[a]HCQ: hydroxychloroquine

[b]SOC: standard of care

[c]central temperature

≥38˚C; [d]NA: not available

SOC groups, respectively, had undetected virus ($p = 0.36$) (Table 2). The analysis of the area under the curve of the Ct value in the study interval showed that the least square mean (SD) was 501.7 (18.0) for the HCQ group and 496.6 (21.2) for the SOC group. The treatment difference was 5.1 (95% CI; -37.1, 47.2) ($p = 0.81$) (Table 3). The revolution of the mean Ct value in the study interval was also studied. The treatment effect on the Ct value was not significant ($p = 0.4717$) (Fig 3). For subjects presenting with mild illness, the median times to negative

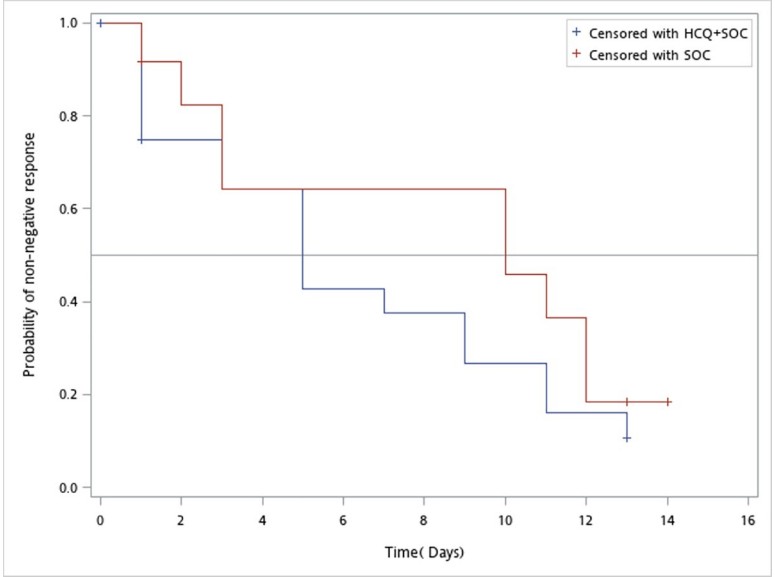

**Fig 2. Probabilities of non-negative responses vs. time (days) for subjects in the HCQ and SOC groups in the multicenter, open-label, randomized controlled trial.** Abbreviations: HCQ: hydroxychloroquine; SOC: standard of care.

rRT-PCR assessment from randomization were 5 days (95% CI; 1, 11 days) for the HCQ group and 11 days (95% CI; 1, 12 days) for the SOC group ($p$ = 0.31) (S1 Fig; S1 Table).

By day 14, 28.6% of the subjects in the HCQ group and 41.7% of the subjects in the SOC group presented with clinical recovery ($p$ = 0.51) (S2 Fig; S2 Table). By day 14, 19.0% and 16.7% of the subjects in the HCQ and SOC groups, respectively, were no longer quarantined. There was no mortality in the present study.

No severe adverse events were reported in the clinical trial. Grades 1 and 2 HCQ-related adverse events included headache (21.1%), dizziness (5.3%), gastritis (5.3%), diarrhea (5.3%), nausea (5.3%), and photophobia (5.3%). The median QTc (ranges) were 429.5 msec (340–467) on day 4 and 421 msec (391–462) on day 8. No severe prolongation was noted.

### Retrospective observational study

Thirty-seven cases were enrolled in the observational study (Fig 1). The mean age (SD) of the subjects was 35.8 (14.5) years. There were 17 (45.9%) male subjects. Twenty-three (82.1%) in

**Table 2. Proportions of negative rRT-PCR assessments on day 14 and median times to negative rRT-PCR results after randomization in the multicenter, open-label, randomized controlled trial.**

| Group | N | Negative[a] | P-value[b] | Median time to negative[c] (Days, 95% CI[d]) | *p*-value[e] |
|---|---|---|---|---|---|
| HCQ[f] | 21 | 17 (81.0%) | 0.71 | 5 (1, 9) | 0.40 |
| SOC[g] | 12 | 9 (75.0%) | | 10 (2, 12) | |

[a]Negative event: both pharyngeal swab and sputum showed negative results

[b]CMH test: stratified by clinical syndrome

[c]Time to negative = Event date or censored date–start day

[d]CI: confidence interval

[e]Log-rank test stratified by clinical syndromes

[f]HCQ: hydroxychloroquine

[g]SOC: standard of care.

**Table 3. Comparison of area under curve of Ct value between subjects in the HCQ and SOC groups in the multicenter, open-label, randomized controlled trial.**

| | HCQ[a] (N = 21) | | | SOC[b] (N = 12) | | | Treatment difference (95% CI)[d] | Treatment effect, *p*-value[e] |
|---|---|---|---|---|---|---|---|---|
| | n | Mean (SD)[c] | Median (range) | n | Mean (SD) | Median (range) | | |
| Observed data | 19 | 506.5 (43.7) | 495.2 (424.0, 608.5) | 11 | 501.6 (67.4) | 495.1 (386.6, 594.5) | 4.9 (-36.5, 46.3) | |
| Least square mean (SE)[f] | | 501.7 (18.0) | | | 496.6 (21.1) | | 5.1 (-37.1, 47.2) | 0.81 |

[a]HCQ: hydroxychloroquine

[b]SOC: standard of care

[c]SD: standard deviation

[d]CI: confidence interval

[e]*p*-value: ANCOVA model

[f]Least-square means based on ANCOVA model with treatment effect. SE is the standard error corresponding to the LSmean.

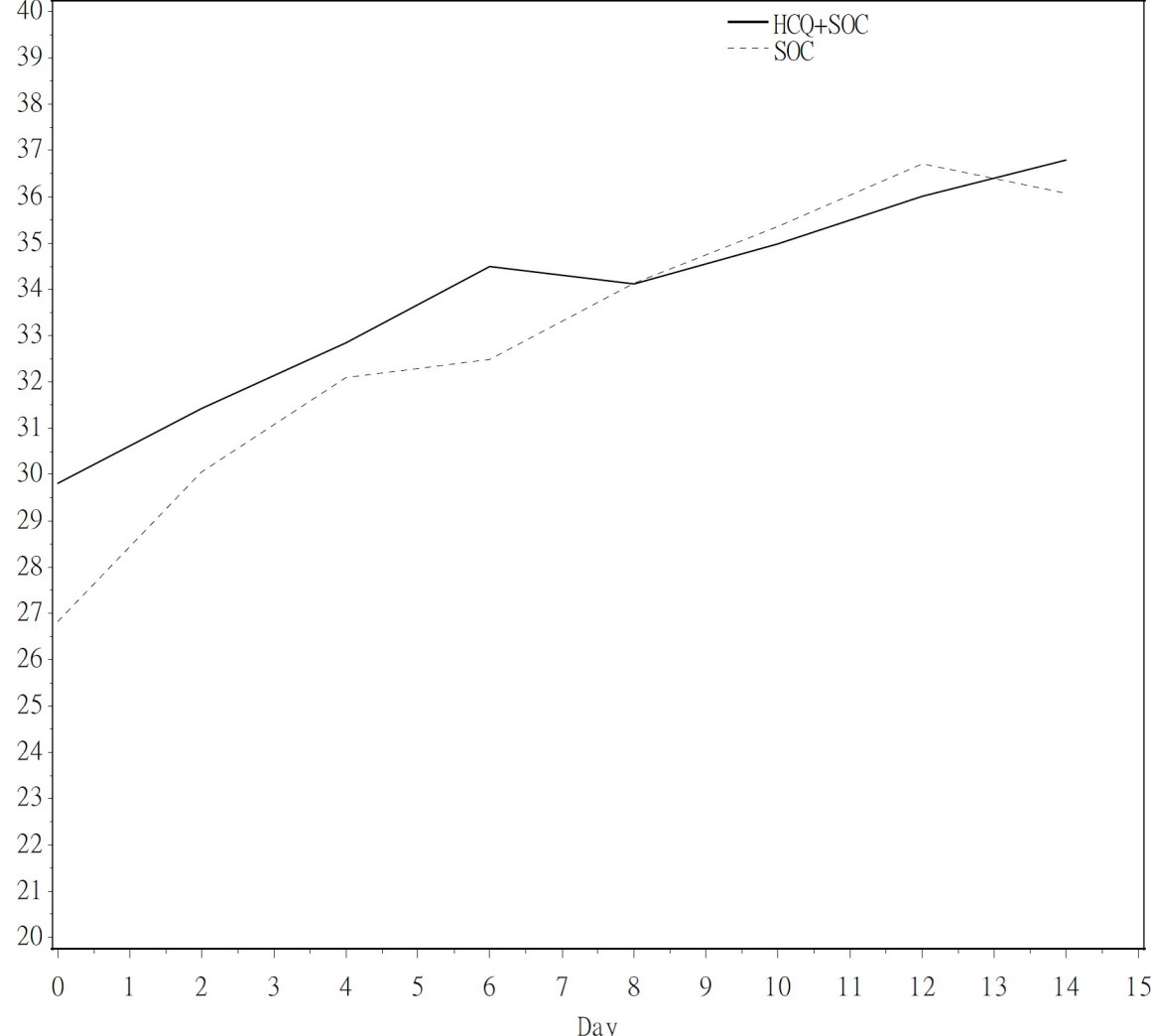

**Fig 3. Mean Ct value of viral rRT-PCR vs. time (days) for subjects in the HCQ and SOC groups in the multicenter, open-label, randomized controlled trial.** Abbreviations: Ct: Cycle threshold; HCQ: hydroxychloroquine; rRT-PCR: real-time reverse transcription polymerase chain reaction; SOC: standard of care.

the HCQ group and zero (0%) in the control group were administered azithromycin concomitantly.

The median times (ranges) to undetected virus were 15 (6–31) days for the HCQ group and 14 (7–22) days for the control group ($p$ = 0.37) (S3 Table). On hospital day 14, the airway samples of 12 subjects (42.9%) in the HCQ group and 5 subjects (55.6%) in the control group turned negative in rRT-PCR ($p$ = 0.70). On hospital day 14, the mean log change (SD) of the Ct value was 7.6 (4.8) in the HCQ group and 11.6 (5.6) in the control group, respectively ($p$ = 0.0625).

## Discussion

The present multicenter, randomized, open-label clinical trial showed that HCQ failed the primary endpoint of shortening the viral clearance interval. The retrospective study also demonstrated that HCQ conferred no therapeutic benefit to the COVID-19 cases investigated here.

Currently, there are > 1,000 ongoing COVID-19 clinical trials worldwide. In multicenter clinical trials conducted in China, chloroquine phosphate has demonstrated efficacy at preventing the progression of COVID-19-related pneumonia [25]. The Chinese guideline [26] recommendation for adults aged 18–65 y is 500 mg twice daily for 7 days in patients weighing > 50 kg and 500 mg twice daily for 2 days followed by 500 mg once daily for 5 days in patients weighing < 50 kg. A clinical trial in Italy planned to include 440 patients and test two different chloroquine doses but was suspended after 81 patients had been enrolled because of excessive QTc prolongation and high mortality rates in the high-dose (600 mg twice daily for 10 days) group [27]. Compared with the study of Borba et al. [27], the participants in our clinical trial were younger, did not receive high HCQ doses, and presented with a low incidence of QTc prolongation.

The first open-label, non-randomized study of HCQ treatment for COVID-19 was conducted in France [15]. Gautret et al. treated 20 patients with 200 mg HCQ thrice daily for 10 days. Six of these patients were administered concomitant azithromycin and 16 other patients received no HCQ therapy. The efficacy of HCQ at clearing the virus was remarkable: 70.0% by day 6 post-inclusion in treated patients vs. 12.5% at day 6 post-inclusion in untreated patients ($p$ < 0.001). However, six of the patients being administered HCQ became clinically worse or were lost to follow-up. Consequently, they were excluded from the final analysis and interpretation of the data became very difficult. Hence, the same team performed an uncontrolled non-comparative observational study on a cohort comprising 80 patients presenting with mild COVID-19 symptoms who underwent HCQ and azithromycin treatment [28]. Subsequently, rapid declines in nasopharyngeal viral load were reported (83% and 93% of the treated patients at days 7 and 8, respectively). A large-scale observational study was conducted on 1,376 COVID-19 patients in New York [29] of whom 58.9% were administered HCQ 600 mg twice on day 1 and 400 mg daily thereafter for a median of 5 days. However, HCQ administration was not associated with the composite intubation or death endpoint (hazard ratio = 1.04; 95% CI = 0.82, 1.32). Another large-scale observational study, conducted on 2,541 COVID-19 patients in southeast Michigan [30], demonstrated that HCQ administration provides a 66% hazard ratio reduction in in-hospital mortality compared with no HCQ treatment ($p$ < 0.001).

A pilot randomized controlled clinical trial (No. NCT04261517) on HCQ therapy for COVID-19 was performed at a single center in Shanghai. It enrolled 30 patients with 1:1 randomization [31]. The study did not reveal any significant difference between the two treatment groups. The viral clearance rates in the throat swab samples were relatively high by day 7 after enrolment in both groups (83.7% vs. 96.3%, respectively; $p$ > 0.05). Moreover, the HCQ dose was comparatively low and the treatment interval was relatively short (400 mg daily for 5 days; no loading). All patients in this trial received aerosolized interferon alpha and most of them

were also administered antiviral drugs that may have diminished or augmented the therapeutic efficacy of HCQ.

Another clinical trial (No. ChiCTR2000029559) enrolled 62 subjects of whom 31 received HCQ 400 mg/d for 5 days. The remaining 31 constituted the control group [32]. After 5 days, the clinical recovery time of the HCQ group was significantly shortened and fever and cough were alleviated relatively faster ($p < 0.05$). Pneumonia improved in 81% of the subjects in the HCQ group and 55% of the patients in the control group ($p < 0.05$). Although this study corroborated the therapeutic efficacy of HCQ, it did not measure or report viral clearance rates.

A recent multicenter, randomized, open-label clinical trial was conducted at 16 COVID-19 treatment centers in China and disclosed negative conversion rates of 85.4% and 81.3% for SARS-CoV-2 28 days after randomization into HCQ + SOC and SOC groups, respectively [33]. In Tang's study, 98.6% (148/150) of the enrolled subjects were categorized as presenting with mild to moderate illness but 63% of the enrollees had also been treated with antiviral agents (arbidol, ribavirin, lopinavir/ritonavir, oseltamivir, or entecavir) which may have interfered with HCQ efficacy.

The strength of our RCT lies in the fact that the enrollees were randomized within 4 days of diagnosis. Thus, the earliest possible intervention could be made. Clinical courses could be clearly and accurately monitored because of early diagnosis and treatment with SOC or HCQ. Furthermore, the HCQ treatment regimen used here consisted of loading twice with 400 mg followed by 200 mg twice daily for 7 days [34]. A physiologically based pharmacokinetic model indicated that the aforementioned dose used in the present study was ideal for HCQ therapy [35]. Third, no antiviral therapy was administered and only one case in the HCQ group and two cases in the SOC group had azithromycin treatment.

In contrast, there were limitations to the present study. Only patients presenting with mild to moderate disease symptoms were enrolled to determine the viral clearance efficacy of HCQ. Hence, there is relatively little data on the impact of severe disease in terms of intubation and mortality. At clinical trial launch, there were no indigenous and very few imported cases in Taiwan. Therefore, study enrollment was prematurely stopped. The low case numbers in the present study might account for the apparent lack of superior efficacy of HCQ, and only reflects the effects of HCQ in young COVID-19 patients with mild symptoms. However, 81% of the HCQ group and 75% of the SOC group had confirmed viral clearance on hospital day 14. Moreover, according to Taiwan CDC regulations, subjects could not leave quarantine until they presented with at least three consecutive negative rRT-PCR results. The outcome of this RCT may assist the Taiwan CDC in its decision to release quarantined patients when medical resources are in short supply. Another limitation of the study was that the mean patient age (SD) was 32.9 (10.7) y as opposed to 51.1 y (13.9) for Borba et al., 45.1 y for Gautret et al., 44.7 y (15.3) for Chen et al., 46.1 y (14.7) for Tang et al, and 63.7 (16.5) for Arshad et al. [15,27,30–32]. For this reason, the observed rates of cardiac and retinal toxicity were low in the present study. Electrocardiogram monitoring was performed frequently and close attention was paid to any changes in patient QTc interval, vision, and neurological symptoms. Lastly, readers might question the gap in median time to negative viral detection between the retrospective observational study and RCT. As the time from infection to admission, and the time from diagnosis to treatment were diverse, and the frequency of viral sampling was not the same in the retrospective observational cohort, the different outcome compared with the present RCT was not surprising.

## Conclusions

Both the retrospective and randomized clinical studies performed here failed to demonstrate HCQ efficacy at shortening viral shedding in the majority of subjects presenting with mild

COVID-19 symptoms. As the case number is small, the absence of evidence is not necessarily evidence of absence and, in future research, large-scale studies involving more patients should be conducted to investigate new agents or combinational therapy and explore viral dynamics.

## Supporting information

**S1 Checklist. CONSORT 2010 checklist of information to include when reporting a randomised trial***.
(DOC)

**S1 Fig. Probabilities of non-negative responses with mild symptoms vs. time (days) for subjects in the HCQ and SOC groups in the multicenter, open-label, randomized controlled trial.** Abbreviations: HCQ: hydroxychloroquine; SOC: standard of care.
(TIF)

**S2 Fig. Probabilities of non-clinical recovery vs. time (days) for subjects in the HCQ and SOC groups in the multicenter, open-label, randomized controlled trial.** Abbreviations: HCQ: hydroxychloroquine; SOC: standard of care.
(TIF)

**S1 Table. Comparison of median times and negative viral rRT-PCR results between subjects in the HCQ and SOC groups presenting with mild symptoms in the multicenter, open-label, randomized controlled trial.**
(DOCX)

**S2 Table. Comparison of times to clinical recovery between subjects in the HCQ and SOC groups in the multicenter, open-label, randomized controlled trial.**
(DOCX)

**S3 Table. Proportions of negative rRT-PCR assessments on day 14 and median times to negative rRT-PCR results in the multicenter, retrospective study.**
(DOCX)

**S1 File.**
(PDF)

## Acknowledgments

The authors thank the National Health Research Institutes, Taiwan Centers for Disease Control, Taiwan Food and Drug Administration, and Center for Drug Evaluation Taiwan for their technical assistance, and all study members participating in the Taiwan HCQ Study Group led by Shu-Hsing Cheng (shcheng@mail.tygh.gov.tw), namely, Chien-Yu Cheng*, Yi-Chun Lin*, Cheng-Pin Chen*, and Shu-Hsing Cheng* from Taoyuan General Hospital, Ministry of Health and Welfare, Chin-Feng Lin, Jiing-Chyuan Luo, Fu-Shun Tsai, Tsung-Yen Yang, Wen-Chen Yau, and Hon-Lai Wong* from Keelung Hospital, Ministry of Health and Welfare, Wu-Pu Lin*, Lin-Chen Chien, Chen-Han Yiu, Chien-Yu Huang, and Yung-Tsung Hsiao from Taipei Hospital, Ministry of Health and Welfare, Ming-Huei Lee and Sz-Rung Huang* from Miaoli General Hospital, Ministry of Health and Welfare, Wei-Sheng Chung*, Tsung-Chia Chen*, and Ting-Yu Tseng* from Taichung Hospital, Ministry of Health and Welfare, Wei-Yao Wang*, Yih-Farng Liou, and Chen-Feng Chiu from Feng Yuan Hospital, Ministry of Health and Welfare, Yi-Wen Huang*, Yang-Hao Yu, Tse-Hung Lin*, and Tz-Yan Chang* from Chang Hua Hospital, Ministry of Health and Welfare, Hung-Chang Hung, Tzung-Fan Chuang, Jia-Hung Liao*, Li-Yueh Yeh, and Shu-Ming Huang from Nantou Hospital, Ministry

of Health and Welfare, Yuan-Der Huang, Shih-Tien Chen, Chi-Min Shin, and Chung-Shin Liao* from Chia Yi Hospital, Ministry of Health and Welfare, Yuan-Pin Hung*, Chih-I Lee, and Chun-Wei Chiu from Tainan Hospital, Ministry of Health and Welfare, and Shah-Hwa Chou, Cheng-Yu Kuo*, Tz-Lun Hung, and Hsin-Hui Wang from Pingtung Hospital, Ministry of Health and Welfare, and Chin-Fu Hsiao from Institute of Population Health Sciences, National Health Research Institutes, Zhunan, Taiwan. *Authorships.

## Author Contributions

**Conceptualization:** Cheng-Pin Chen, Yi-Chun Lin, Chien-Yu Cheng, Shu-Hsing Cheng.

**Data curation:** Yi-Chun Lin, Chin-Fu Hsiao, Shu-Hsing Cheng.

**Formal analysis:** Cheng-Pin Chen, Yi-Chun Lin, Chin-Fu Hsiao, Chien-Yu Cheng, Shu-Hsing Cheng.

**Funding acquisition:** Shu-Hsing Cheng.

**Investigation:** Cheng-Pin Chen, Tsung-Chia Chen, Ting-Yu Tseng, Hon-Lai Wong, Cheng-Yu Kuo, Wu-Pu Lin, Sz-Rung Huang, Wei-Yao Wang, Jia-Hung Liao, Chung-Shin Liao, Yuan-Pin Hung, Tse-Hung Lin, Tz-Yan Chang, Yi-Wen Huang, Wei-Sheng Chung, Chien-Yu Cheng.

**Methodology:** Chin-Fu Hsiao, Chien-Yu Cheng.

**Supervision:** Yi-Wen Huang, Wei-Sheng Chung.

**Writing – original draft:** Cheng-Pin Chen, Yi-Chun Lin.

**Writing – review & editing:** Chien-Yu Cheng, Shu-Hsing Cheng.

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
