## [Decision Letter · Decision Letter 0]

8 Sep 2020

PONE-D-20-20024

A Multicenter, randomized, open-label, controlled trial to evaluate the efficacy and tolerability of hydroxychloroquine and a retrospective study in adult patients with mild to moderate Coronavirus disease 2019 (COVID-19)

PLOS ONE

Dear Dr. Cheng,

Thank you for submitting your manuscript to PLOS ONE. After careful consideration, we feel that it has merit but does not fully meet PLOS ONE’s publication criteria as it currently stands. Therefore, we invite you to submit a revised version of the manuscript that addresses the points raised during the review process.

The reviewers have issues regarding the statistical methods and study methodology that need to be addressed

We look forward to receiving your revised manuscript.

Kind regards,

Stephen L Atkin, MD

Academic Editor

PLOS ONE

Journal Requirements:

2. We note that items are missing from your CONSORT flowchart in table 1. Please revise your flow chart according to this sample diagram: http://www.consort-statement.org/consort-statement/flow-diagram.

3. Thank you for submitting your clinical trial to PLOS ONE and for providing the name of the registry and the registration number. The information in the registry entry suggests that your trial was registered after patient recruitment began. PLOS ONE strongly encourages authors to register all trials before recruiting the first participant in a study.As per the journal’s editorial policy, please include in the Methods section of your paper:

1) your reasons for your delay in registering this study (after enrolment of participants started);

2) confirmation that all related trials are registered by stating: “The authors confirm that all ongoing and related trials for this drug/intervention are registered”.

"The authors thank the Hospital and Social Welfare Organizations Administration Commission,

 Ministry of Health and Welfare for their research grant. This funding source played no role in

study design or conduction, data collection, analysis or interpretation, writing of the

manuscript, or decision to submit it for publication. The authors also thank Taiwan Biotech

Co. Ltd. for their donation of investigational products..."

"SHC received research grant from the Hospital and Social Welfare Organizations Administration Commission, Ministry of Health and Welfare (https://www.hso.mohw.gov.tw/). This funding source played no role in study design or conduction, data collection, analysis or interpretation, writing of the manuscript, or decision to submit it for publication. "

Additionally, because some of your funding information pertains to commercial funding, we ask you to provide an updated Competing Interests statement, declaring all sources of commercial funding.

In your Competing Interests statement, please confirm that your commercial funding does not alter your adherence to PLOS ONE Editorial policies and criteria by including the following statement: "This does not alter our adherence to PLOS ONE policies on sharing data and materials.” as detailed online in our guide for authors  http://journals.plos.org/plosone/s/competing-interests.  If this statement is not true and your adherence to PLOS policies on sharing data and materials is altered, please explain how.

Please include the updated Competing Interests Statement and Funding Statement in your cover letter. We will change the online submission form on your behalf.

6. Please include captions for figure 3 and 4.

7. Please include a copy of Tables 3, 4, 5 which you refer to in your text on page 12.

Reviewers' comments:

Reviewer's Responses to Questions

**Comments to the Author**

1. Is the manuscript technically sound, and do the data support the conclusions?

Reviewer #1: Yes

Reviewer #2: Yes

2. Has the statistical analysis been performed appropriately and rigorously? 

Reviewer #1: Yes

Reviewer #2: Yes

3. Have the authors made all data underlying the findings in their manuscript fully available?

Reviewer #1: Yes

Reviewer #2: No

4. Is the manuscript presented in an intelligible fashion and written in standard English?

Reviewer #1: Yes

Reviewer #2: Yes

5. Review Comments to the Author

Reviewer #1: The authors present data from a Taiwan-based multi-center (11 publically funded hospitals) studies addressing the question of whether hydroxychloroquine has therapeutic efficacy for the treatment of COVID-19. Data is presented for an open label RCT (NCT04384380) and an observational retrospective study conducted for up to 14 days. There are limitations to the study that to a greater extent the authors have emphasized but a number of issues need to better emphasized and discussed.

1/ The authors have explained that recruitment was limited to few cases presenting in Taiwan. Only 33 were included for the RCT and 37 for the retrospective study. These numbers are very small. Furthermore, as can be seen from Table 1 there are very few who are classified as “moderate” – 2 for HCQ and 2 for SOC in the RCT and 5 and 3 respectively in the retrospective study. This needs to be better emphasized as the study data and conclusions really can only relate to mild cases AND comparatively also a young population ~30-35. Discussion could be added at lines 384/5 that due to few numbers the RCT data really only reflects that from young COVID-19 patients with mild symptoms.

2/ According to the study design lines 196-197 only those with moderate symptoms received antimicrobial treatment. Correct? If correct, emphasise that patients with mild symptoms only received HCQ or SOC.

2/ A clearer description of symptoms would be beneficial. For instance:

i Cough? How was this defined? The majority of the patients are listed as presenting with cough. Is this persistent cough, or --?

ii. Fever? How was this defined? Very few – one only had fever in the HCQ RCT.

3. PCR data for viral load was obtained every other day. Is it possible to compare the rates of decline of viral load for the RCT study HCQ vs SOC?

4. The authors have discussed their results in comparison with other published studies. I suggest adding comparison with the data from a large retrospective study from the Henry Ford Hospital in Detroit https://doi.org/10.1016/j.ijid.2020.06.099 published in the International J of Infectious Diseases (Dr Samia Arshad) et al that concludes that HCQ is effective and reduces mortality. However, note that the greater % of patients were also prescribed (cortico)steroids and median age was 64.

Reviewer #2: I would very much hope that the authors would consider making their data available in a de-identified manner in a simple spreadsheet readable format like a comma separated variable file that could be readily ingested into a stats package. I have made comments in a track changed format in the attachment. The only additional query I would raise is that the authors need to acknowledge the small sample sizes of the two study cohorts they have reported on and to maybe also acknowledge that absence of evidence is not necessarily evidence of absence and that this is especially the case in small studies.

6. PLOS authors have the option to publish the peer review history of their article (what does this mean?). If published, this will include your full peer review and any attached files.

Reviewer #1: **Yes: **Chris R Triggle

Reviewer #2: **Yes: **Greg Fegan

---

## [Author Response · Author response to Decision Letter 0]

15 Oct 2020

To the editors and reviewers, 

This rebuttal letter responds to each point raised by the academic editor and reviewers. The authors have prepared a marked-up copy that highlights the changes made to the original version, and also an unmarked version.

The authors have deposited the protocols in protocols.io. DOI: dx.doi.org/10.17504/protocol.io.bmfak3ie

The authors have ensured that the manuscript meets PLOS ONE's style requirements, including those for file naming. The item missed from the CONSORT flowchart was amended. 

The authors understand the importance of trial registration before recruiting the first participant in a study. The reasons for the delay in registering this study (after enrolment of participants started) were that the appendix of the protocol was in Mandarin and had to be translated to English to meet QC criteria; the registration of the clinical trial was accepted on May/12/2020, after the first enrollment on Apr/1/2020. However, the authors confirm that all ongoing and related trials for this drug/intervention are registered. This statement has been added under the subheading “Study design.”

In the cover level, the corresponding author stated that she received research funding from the Hospital and Social Welfare Organizations Administration Commission, Ministry of Health and Welfare, Taiwan Biotech Co. Ltd. for their donation of investigational products. This funding source played no role in study design or conduction, data collection, analysis or interpretation, writing of the manuscript, or decision to submit it for publication. Furthermore, the commercial funding did not alter our adherence to PLOS ONE policies on the sharing of data and materials.

As there are no legal or ethical concerns, the authors have shared the data publicly on figshare (DOI: https://doi.org/10.6084/m9.figshare.12631403.v1). 

Captions for Figure 3 and 4 have been added. Tables 3, 4, and 5 have also been added. Supporting information is provided at the end of the manuscript and the in-text citations have been updated accordingly. 

Reviewer #1: 

The authors present data from a Taiwan-based multi-center (11 publically funded hospitals) studies addressing the question of whether hydroxychloroquine has therapeutic efficacy for the treatment of COVID-19. Data is presented for an open label RCT (NCT04384380) and an observational retrospective study conducted for up to 14 days. There are limitations to the study that to a greater extent the authors have emphasized but a number of issues need to better emphasized and discussed.

1) The authors have explained that recruitment was limited to few cases presenting in Taiwan. Only 33 were included for the RCT and 37 for the retrospective study. These numbers are very small. Furthermore, as can be seen from Table 1 there are very few who are classified as “moderate” – 2 for HCQ and 2 for SOC in the RCT and 5 and 3 respectively in the retrospective study. This needs to be better emphasized as the study data and conclusions really can only relate to mild cases AND comparatively also a young population ~30-35. Discussion could be added at lines 384/5 that due to few numbers the RCT data really only reflects that from young COVID-19 patients with mild symptoms.

Response: The authors have emphasized the limitations of the young population and mild illness in the “Results (line 255-259 of Page 13), “Discussion” (lines 412-418 of page 23), and “Conclusion” (line 438-439 of page 25). 

2) According to the study design lines 196-197 only those with moderate symptoms received antimicrobial treatment. Correct? If correct, emphasise that patients with mild symptoms only received HCQ or SOC. 

Response: The subjects with mild symptoms did not receive antimicrobial treatment. The authors have clarified this in “Study design” (lines 196-198 of page 10).

3) A clearer description of symptoms would be beneficial. For instance:

i Cough? How was this defined? The majority of the patients are listed as presenting with cough. Is this persistent cough, or --?

ii. Fever? How was this defined? Very few – one only had fever in the HCQ RCT. 

Response: 1. The severity of cough was defined as mild (no symptomatic treatment), moderate (responsive to symptomatic treatment), and severe (symptomatic treatment given but without response). Most subjects had mild cough and the authors have added the data in “Results” (line 257 of page 13). 2. A central temperature of more than 38 °C is defined as fever (Table 1). 

4) PCR data for viral load was obtained every other day. Is it possible to compare the rates of decline of viral load for the RCT study HCQ vs SOC?

Response: The rates of decline are presented in Fig. 3 and the difference was not significant between the groups. 

5) The authors have discussed their results in comparison with other published studies. I suggest adding comparison with the data from a large retrospective study from the Henry Ford Hospital in Detroit https://doi.org/10.1016/j.ijid.2020.06.099 published in the International J of Infectious Diseases (Dr Samia Arshad) et al that concludes that HCQ is effective and reduces mortality. However, note that the greater % of patients were also prescribed (cortico)steroids and median age was 64.

Response: The retrospective study of Arshad et al. from Henry Ford Hospital was cited and discussed (lines 372-375 of page 20 and line 425-426 of page 22).

Reviewer #2: I would very much hope that the authors would consider making their data available in a de-identified manner in a simple spreadsheet readable format like a comma separated variable file that could be readily ingested into a stats package. I have made comments in a track changed format in the attachment. The only additional query I would raise is that the authors need to acknowledge the small sample sizes of the two study cohorts they have reported on and to maybe also acknowledge that absence of evidence is not necessarily evidence of absence and that this is especially the case in small studies.

Response: 

1) In figshare (DOI: https://doi.org/10.6084/m9.figshare.12631403.v1), the authors have provided a simple spreadsheet readable format that can be readily ingested into a stats package. 

2) The authors acknowledge that “Both the retrospective and randomized clinical studies performed here failed to demonstrate HCQ efficacy at shortening viral shedding in the majority of subjects presenting with mild COVID-19 symptoms. As the case number is small, absence of evidence is not necessarily evidence of absence…” and have added this statement in the “Conclusions” (lines 439-430 of pages 23 ). 

3) The comments in the “track changed format” were replied as follows: 

i. What’s wrong with X-ray? 

Response: The authors would like to change them. 

ii. These adjacent statements seem contradictory. 

Response: The authors have clarified that cases of severe illness were not enrolled (lines 159-160 of page 8).

iii. Is this simply related to study size given the allocation ratio?

Response: The estimated sample size was 45 (Lane 194 of page 10).

iv. Tense needs changing

Response: Thank you. We have changed a number of phrases in the Methods from the future to the past and present tense, as appropriate. 

v. Suggest you rephrase to the trial was prospectively/retrospectively (delete accordingly) registered at (give URL) 

Response: The authors have stated that “this prospective randomized controlled clinical trial was registered at https://clinicaltrials.gov/ct2/show/NCT04384380?term=Hydroxychloroquine&cond=COVID-19&cntry=TW&draw=2&rank=1(NCT04384380). (lines 205-207 of page 10). 

vi. bit picky - but no hospitals are mentioned just geographic locales across Taiwan.

Response: We have clarified that the study was conducted at the same 11 hospitals (line 148-151 of page 8).

vii. Suggest you combine this into 3 columns eg mean and (sd) in one cell 

Response: Table 1 has been adjusted to use only three columns. 

viii. I take it that the 95% CI is (-37.1, 47.2) This needs clarifying. Would suggest you report all CIs as LB,UB not LB-UB) as you do in Table below

Response: Your interpretation was correct. The data have been adjusted.

---

## [Decision Letter · Decision Letter 1]

10 Nov 2020

A multicenter, randomized, open-label, controlled trial to evaluate the efficacy and tolerability of hydroxychloroquine and a retrospective study in adult patients with mild to moderate Coronavirus disease 2019 (COVID-19)

PONE-D-20-20024R1

Dear Dr. Cheng,

We’re pleased to inform you that your manuscript has been judged scientifically suitable for publication and will be formally accepted for publication once it meets all outstanding technical requirements.

Kind regards,

Stephen L Atkin, MD

Academic Editor

PLOS ONE

Additional Editor Comments (optional):

Reviewers' comments:

Reviewer's Responses to Questions

**Comments to the Author**

1. If the authors have adequately addressed your comments raised in a previous round of review and you feel that this manuscript is now acceptable for publication, you may indicate that here to bypass the “Comments to the Author” section, enter your conflict of interest statement in the “Confidential to Editor” section, and submit your "Accept" recommendation.

Reviewer #2: All comments have been addressed

2. Is the manuscript technically sound, and do the data support the conclusions?

Reviewer #2: (No Response)

3. Has the statistical analysis been performed appropriately and rigorously? 

Reviewer #2: (No Response)

4. Have the authors made all data underlying the findings in their manuscript fully available?

Reviewer #2: (No Response)

5. Is the manuscript presented in an intelligible fashion and written in standard English?

Reviewer #2: (No Response)

6. Review Comments to the Author

Reviewer #2: (No Response)

7. PLOS authors have the option to publish the peer review history of their article (what does this mean?). If published, this will include your full peer review and any attached files.

Reviewer #2: **Yes: **Greg Fegan

---

## [Editor Report · Acceptance letter]

18 Nov 2020

PONE-D-20-20024R1 

A multicenter, randomized, open-label, controlled trial to evaluate the efficacy and tolerability of hydroxychloroquine and a retrospective study in adult patients with mild to moderate coronavirus disease 2019 (COVID-19) 

Dear Dr. Cheng:

I'm pleased to inform you that your manuscript has been deemed suitable for publication in PLOS ONE. Congratulations! Your manuscript is now with our production department. 

Kind regards, 

on behalf of

Dr. Stephen L Atkin 

Academic Editor

PLOS ONE